# Wood Chip Storage in Small Scale Piles as a Tool to Eliminate Selected Risks

**Miloš Gejdoš *** and **Martin Lieskovský**

Department of Forest Harvesting, Logistics and Ameliorations, Faculty of Forestry,
Technical University in Zvolen, T.G. Masaryka 24, 96001 Zvolen, Slovakia; lieskovsky@tuzvo.sk
* Correspondence: gejdos@tuzvo.sk; Tel.: +421-455-206-283

**Abstract:** Massive use of wood biomass is usually associated with its long-term, large-scale storage in power plants and heating plants. Long-term storage of wood biomass (more than 3 months), in large volumes, brings risks from the point of view of human health or property treatment. This work aimed to verify how the long-term storage of wood chips from beech wood in small piles affects their energy properties and whether in this way it is possible to reduce the risk of fire by self-heating in piles and the volume of phytopathogenic spore production. Four experimental piles, each with a base of $4 \times 4$ m and a height of 2 m, were established. After 6 months, one of the piles was disassembled and samples from 0.5 m, 1.0 m, and 1.5 m height levels were taken for analysis. The results of the experiment confirmed that the energy properties of wood chips stored in small piles significantly deteriorate after more than half a year of storage. It has also been confirmed that the choice of this method of storing in smaller, spatially divided piles can lead to a significant minimization of the risk of spontaneous combustion and fire. The length of the storage period did not have a positive effect on the abundance of phytopathogen content in the stored piles. With the length of storage, only the number of identified harmful species of phytopathogens were changed and, at the same time, their number of colonies increased.

**Keywords:** forest biomass; wood chips; biomass storage; risks; self-heating; phytopathogen

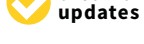



## 1. Introduction

With a growing share of the use of wood-based renewable energy sources throughout the European Union, the issues of their quality, availability, and solutions to logistical problems are becoming more and more important [1–6]. Mass use of wood biomass is usually associated with its long-term, large-scale storage in power plants and heating plants [7–11]. There are a lot of parallels between storing biomass for each of these applications (e.g., chip storage for pulp and paper production) [12–14]. The majority of heat energy suppliers in Slovakia have installed technologies that require high quality wood chips. They focus mainly on the moisture content and calorific value of the fuel, mainly because of their effect on final price. These two basic properties can also be influenced by the method and length of storage [15,16]. In the course of the first year of storage, biomass energy loss can range from 25% to 55%, caused by an increase in the relative moisture and degradation of stored chips [17,18]. Long-term storage of wood biomass (more than 3 months), in large volumes, brings risks from the point of view of human health or property treatment. These risks are mainly connected with the possibility of fire or phytopathogenic spore production, which can seriously damage human health [19–23]. Similarly, long-term storage of wood biomass naturally degrades this biological material and degrades its energy properties [21,24–27].

Several methods to eliminate the risks of fire and deterioration of the energy properties of stored biomass have been developed and described [28–32]. These methods include, in particular, the regular spreading, venting, and covering of large piles of wood chips. Various techniques for storing wood chips have been developed in parallel with research into the risks associated with their storage.

This work aimed to verify how the long-term storage of beech wood chips in small piles affects their energy properties. The aim was also to verify whether the use of this method of storing chips can influence the risk of fire due to spontaneous combustion and also reduce the risks to human health due to the production of spores of phytopathogens and molds.

## 2. Materials and Methods

### 2.1. Material and Experimental Piles

The piles of wood chips designed for the experiment were produced from European beech (*Fagus sylvatica* L.). This tree species was selected mainly because of the tree species composition of the University Forestry Enterprise and its good availability for the implementation of the experiment and because this tree species is very often used in Slovakia as a renewable energy source. Chips from biomass were produced right after the trees were cut down (16 July 2018), in operating conditions of Wood depot storage on University Forestry Enterprise (Lieskovec, Slovakia). Thinner logs (up to 30 cm in medium thickness) were chipped with a Musmax wood terminator 10 Z mobile drum chipper (Manufacturer: Landtechnik Urch GmbH, Oberer Markt, Austria).

Wood chip piles were located in an area of Lieskovec, within the University Forest Enterprise near the town of Zvolen, Slovakia. Four piles were created in the shape of a pyramid, each with a base of 4 m × 4 m and a height of 2 m (Figure 1). Samples were taken from the piles at six month intervals (every six-months, one of the piles was dismantled and samples were taken from inside the pile) at three distances from the ground: 0.5 m, 1.0 m, and 1.5 m. In total, the experiment duration was 2 years. Temperature measurement was carried out using the electronic probe "Pt 100/B", placed at three height levels: 0.5 m, 1.0 m, and 1.5 m in two piles (6 probes). Each probe was connected to the "Datalogger OMR 700"(Manufacturer: Techreg Ltd., Lučenec, Slovakia), serving for storing the data in 10 min intervals for 2 years.

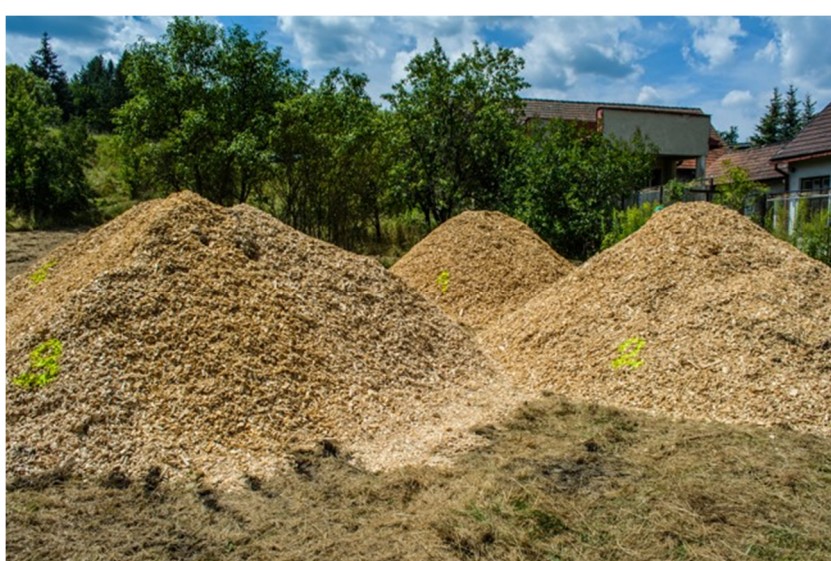

**Figure 1.** Percentage occurrence of recorded qualitative features and their average size in the entire sample of non-coniferous trees.

The development of temperature and relative air humidity was recorded during the whole duration of storage every day in 10 min intervals with the help of a meteorological station (from Environmental Measuring Systems Ltd. Brno, Czech republic) located directly in the area of the storage piles—120 m from the piles. The weather station was installed in the research site approximately 3 months after the establishment of the experiment (19 October 2018).

### 2.2. Laboratory Analysis

Samples were taken from the piles at six months intervals at three distances from the ground: 0.5 m, 1.0 m, and 1.5 m. One of the four experimental piles (Figure 1) was always dismantled for sampling. The samples were stored in airtight sealed plastic bags until performing the analyses. The piles were established on 16 July 2018. Samples were taken on 6 February 2019, 12 August 2019, and 5 February 2020, and the last one was taken on 5 August 2020. Microbiological identification of fungi was performed in the accredited laboratory of the Regional Public Health Authority in Poprad. The samples were stored in airtight sealed plastic bags until performing the analyses. Microbiological identification of fungi was performed by examination, ISO 21527-2 [33]. The quantification of the fungi (Colony forming units per gram (CFU.g$^{-1}$)) was performed in compliance with standard STN 56 0100 [34]. Different species of fungi were identified, in compliance with the scientific literature [35]. For the identification and isolation of the microscopic fungi, cultures created on agar were used. Agar plates were cultivated at a temperature of 25 °C $\pm$ 1 °C for 5 to 7 days. Isolates were inoculated to the cultivating medium. For the identification, the following cultivating media were used: Yeasts Extract Agar (CYA), Malt Extract Agar (MEA), Creatine Sucrose Agar (CREA), and CzapekYeasts Extract Agar with 20% sucrose (CY20S). After obtaining pure microscopic cultures, their identification was performed based on morphological and cultural characteristics according to the guides listed in the literature [36–38].

When studying the colonies, the following features were observed:

- Cultivation features—observed on the identification agars (shape of the colony, size of the colony, speed of colony growth, colony color, production of exudates on the colony surface, production and excretion of pigments into the environment);
- Micromorphologic features—observed in the amounts with lactoferrin (occurrence of asexual spores, their size, shape, method of creation and arrangement; type of the vegetative fructification structure, its arrangement, and shape; the occurrence of special structures; the occurrence of sexual fructification structures and spores, their size, shape, and arrangement).

The results were in the form of occurrence frequency, relative density, and quantitative occurrence of microscopic fungi. The frequency of occurrence (Fr) and relative density (RD) of the species and genera of the microscopic fungi is calculated according to the relations 1 and 2 [39]:

$$Fr = (ns/N).\ 100,\ [\%] \tag{1}$$

$$RD = (ni/NI).\ 100,\ [\%]. \tag{2}$$

Fr—frequency of occurrence;
RD—relative density;
ns—the number of samples in which the genus or species was detected;
N—total number of samples;
NI—total number of isolates;
ni—number of isolates of the given genus or species.

To determine the number of microscopic fungi in 1 g, i.e., to find out the colonies comprising a unit (CFU.g$^{-1}$), which were isolated, Petri dishes from two dilutions, one after another, consisting of a maximum of 150 colonies, were used. To determine the number of microscopic fungi CFU in 1 g, the following relation nr. 3 was used [33]:

$$N = (\textstyle\sum c)/(V.[(n_1 + 0{,}1.n_2).d]),\ [CFU.g^{-1}] \tag{3}$$

$\Sigma$c—the sum of characteristic colonies on selected dishes, used for the calculation;
$n_1$—Number of dishes of 1st dilution used to calculate;
$n_2$—number of dishes of 2nd dilution used to calculate;
d—dilution factor is identical to the 1st dilution used;
V—the volume of inoculum used for the inoculation of cultivating medium.

The quantification of fungi was analyzed according to the standard ISO 21527-2 [33]. The number of CFU was rounded off by the STN EN ISO 7218 [40].

Standard methods were used to measure the various biomass energy properties (Moisture content, Combustion heat, Calorific value, Ash Content).

The samples were also used to determine the moisture content by drying and reweighing the sample [25,26]. The samples were dried at a temperature of 105 °C $\pm$ 2 °C until they reached a constant weight. After reweighing the samples using laboratory scales with an accuracy of 0.01 g, the values of relative moisture content of the wood chips were calculated. The relative moisture content at the individual sampling heights was calculated as the ratio of water weight contained in the samples to the weight of the wet samples, and it was expressed as a percentage. The calorific value of the material (calorific value in MJ/kg) was determined by the bomb calorimeter IKA C200 (Manufacturer: IKA GmbH, Staufen, Germany) using the standard ignition method STN ISO 1928 and ÖNORM M 7132 [41,42], and the ash content was determined using the standard ignition method STN ISO 1171 [43].

Relative moisture content ($w_r$) is defined as the percentage of the amount of water contained in the fuel relative to the measured wet moisture of the fuel (4):

$$w_r = (m_w - m_0)/m_w \cdot 100, [\%] \tag{4}$$

$m_w$—wet fuel sample weight [kg];
$m_0$—weight of fuel sample after drying [kg].

Combustion heat $Q_s$ (MJ/kg) is the heat released by the perfect combustion of 1 kg of fuel to $CO_2$, $SO_2$, and liquid water, $H_2O$.

Calorific value $Q_i$ (MJ/kg) is the heat released under the same conditions, except that steam is released instead of liquid water. The calorific value is calculated from the combustion heat by subtracting the heat of vaporization of the water, $Q_v$. The water released by combustion is the sum of the water contained in the fuel (its moisture) and the water produced by the combustion of the fuel (corresponds to the hydrogen content in the fuel). In many cases, the heat of combustion and calorific value is expressed based on kWh/kg. The following applies to the conversion: 1 kWh/kg = 3.6 MJ/kg.

The chemical composition of wood has a high effect on the calorific value, especially the content of carbon and hydrogen, which form a substantial part of the combustibility in the individual components of wood. Different types of calorimeters are used to determine the heat of combustion and to calculate the calorific value. The measurement of combustion heat of solid fuels was performed according to STN ISO 1928 [41]. The ash content in the woody biomass was determined according to the standard STN EN ISO 18122 [44].

## 3. Results

In an experiment focused on the analysis of the health risks from experimental piles of beech wood chips, the parameters of the occurrence of phytopathogens after a certain storage period were analyzed. At the same time, the degradation of energy properties were evaluated (combustion heat, calorific value (MJ.kg$^{-1}$) according to ÖNORM M 7132 at 0% humidity, calorific value (MJ.kg$^{-1}$) according to ÖNORM M 7132 as delivered, calorific value (MJ.kg$^{-1}$) according to STN ISO 1928 in the delivered condition). Atmospheric temperature and relative humidity data were collected from 19 October 2018. Data on atmospheric temperature and relative humidity were collected from the weather station at ten-minute intervals. Data on the development of temperatures in the piles at height levels of 0.5 m, 1.0 m, and 1.5 m were obtained from temperature probes.

### 3.1. The Development of Atmospheric Temperature and Relative Air Humidity

The monitoring of atmospheric conditions was performed for almost a year and ten months. Figure 2 clearly shows that after lower temperatures and higher relative humidity during the first winter period of the experiment, the atmospheric temperature gradually

started to increase again from February, and the relative humidity decreased. A similar development occurred in the second winter period, but temperatures were higher and humidity at the turn of March and April 2020 reached values below 20%. Thus, atmospheric conditions had a partial effect on the development of phytopathogens at the end of the storage period, where air temperatures regularly exceeded 20 °C and the relative humidity did not fall below 30%. Warm and humid weather prevailed, which differed in part from the previous summer period. This may have partially affected the population of phytopathogens in the stored piles.

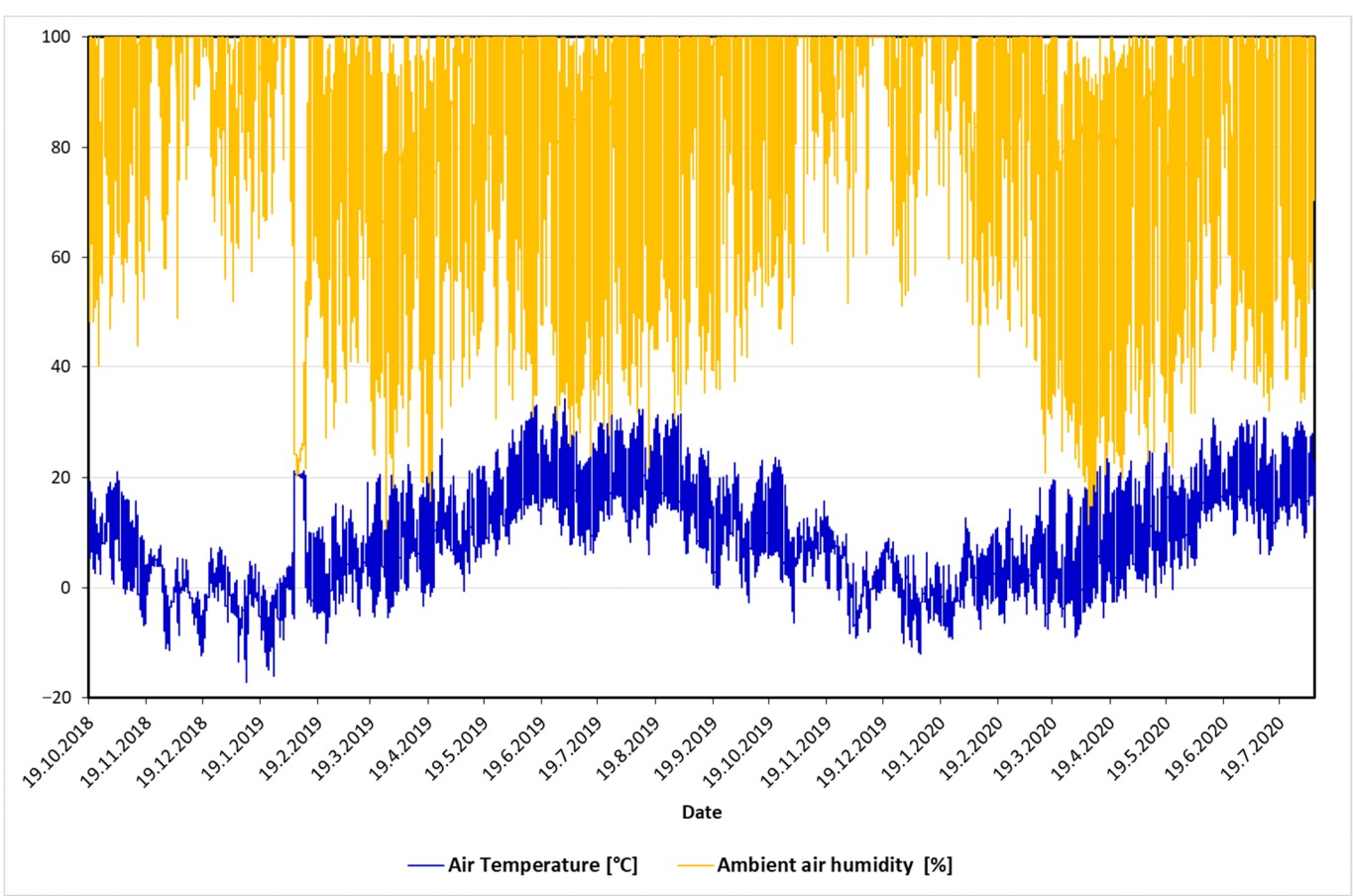

**Figure 2.** Ambient air humidity and air temperature during the storage period.

### 3.2. Heat Development in Experimental Piles

The development of temperatures was monitored in two (T1 and T2) of the four total experimental piles due to economic constraints. The development of temperatures in the piles was accompanied by relatively dry and warm weather during the establishment. For this reason, the activity of the microorganisms in the piles was not so intense and the piles heated much less than is usual with this material. From February 2019, with the increase of atmospheric temperature, the temperatures in the experimental piles also increased (Figure 3). However, the maxima did not exceed 32 °C until the date of dismantling the second pile, which is an extremely interesting result in terms of the risk of spontaneous combustion. In February 2020, the last experimental pile was dismantled, so the measurement was only in the last pile. Since March 2020, in correlation with the atmospheric temperature, temperatures have risen again at all measurement levels of the pile. The highest temperatures were reached in the last monitored pile during the whole period in June and July 2020. The highest temperature levels were again reached in the middle level of the pile. Again, the fundamental influence of atmospheric temperature and its influence on the development of fungal phytopathogens was noticeable. In July 2020, the highest

temperatures during the whole duration of the experiment were recorded in the pile (above 30 °C), which can be attributed directly to the development of atmospheric temperature during this period.

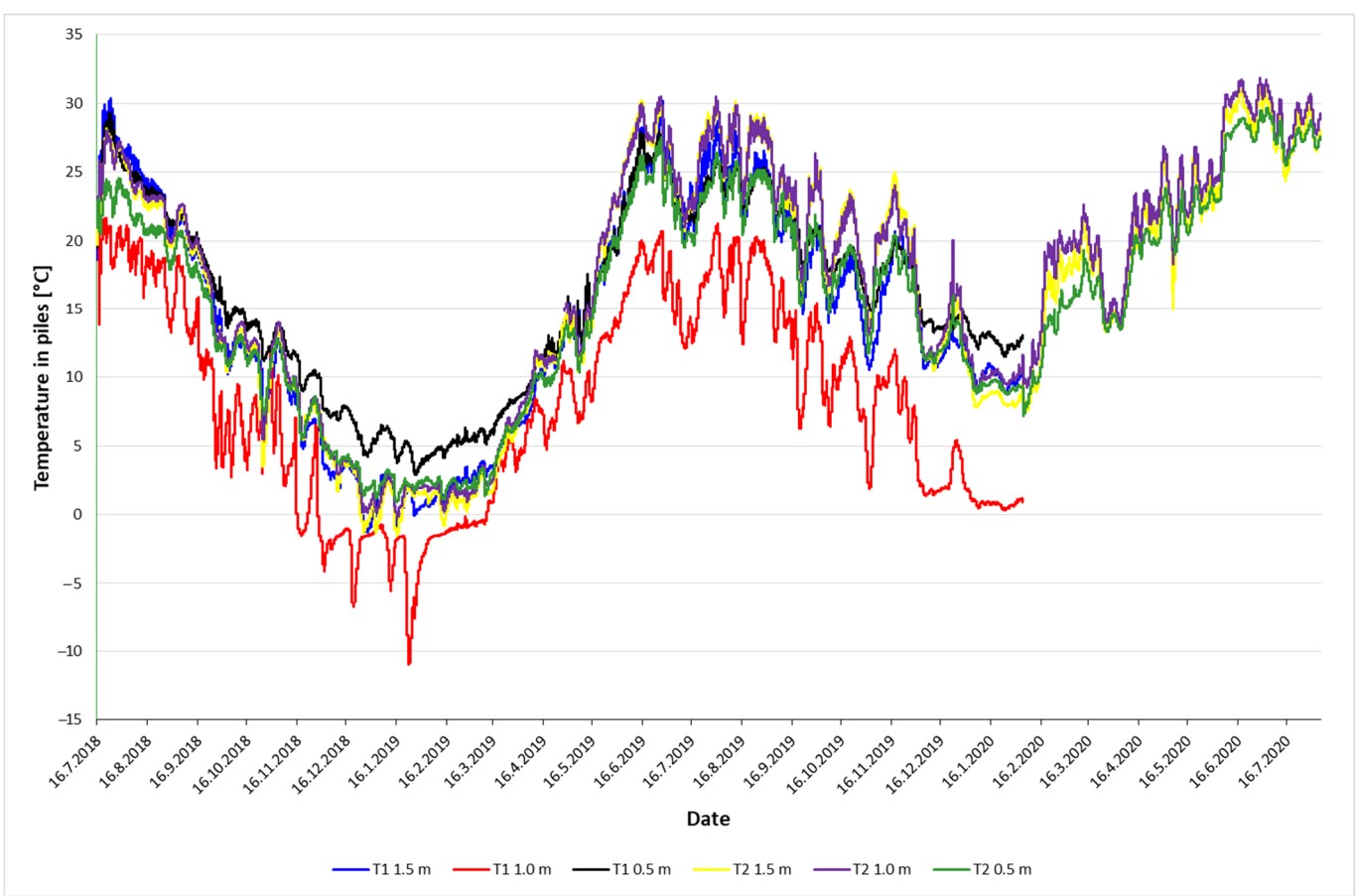

**Figure 3.** The temperature in the experimental piles during the storage period 2018–2020.

The highest levels of temperature in the highest levels of the piles were recorded (1.0–1.5 m), which is related to the increase in air temperature. The lowest temperature levels have long been shown by the probe at a height of 1.0 m in experimental pile no. 1 (T1 1.0 m). At this height level, even in the hottest period, they did not exceed 25 °C. Quite unexpected developments in the piles have confirmed that if the piles are established in smaller dimensions and during periods of warm and dry weather, heating in the pile can be eliminated quite well. However, the activity of phytopathogens cannot be eliminated, and it is these temperature conditions that represent their optimum.

The development of temperatures in small experimental piles can be described as surprising and suggests that such a method of storage can eliminate the serious risk of spontaneous combustion of biomass. However, in addition to the dimensional parameters of the piles, the atmospheric conditions at the time of their establishment also contribute to this, as well as the moisture content parameters of the wood biomass from which the chips were produced and the piles established.

### 3.3. Development of Energy Parameters in Experimental Piles

The development of energy parameters of chips taken from three height levels from the experimental piles is shown in Table 1.

**Table 1.** Overview of energy parameters of wood chips taken from three height levels from the experimental piles.

| Energy Parameter | Sample Height/Date of Dismantling the Pile | | |
| --- | --- | --- | --- |
| | Pile 1 Sampling 6 February 2019 | | |
| | 0.5 m | 1.0 m | 1.5 m |
| Moisture content (%) | 28.1 | 34.4 | 48.4 |
| Combustion heat (MJ.kg$^{-1}$) | 19.335 | 19.175 | 19.222 |
| Calorific value (MJ.kg$^{-1}$) determined by ÖNORM M 7132 by moisture 0% | 17.993 | 17.833 | 17.880 |
| Calorific value (MJ.kg$^{-1}$) determined by ÖNORM M 7132 in delivered condition | 12.257 | 10.869 | 8.036 |
| Calorific value (MJ.kg$^{-1}$) determined by STN ISO 1928 in delivered condition | 12.373 | 10.987 | 8.158 |
| Ash content (%) | 0.60 | 0.51 | 0.54 |
| | Pile 2 Sampling 12 August 2019 | | |
| Moisture content (%) | 39.9 | 49.1 | 55.0 |
| Combustion heat (MJ.kg$^{-1}$) | 19.175 | 19.045 | 19.112 |
| Calorific value (MJ.kg$^{-1}$) determined by ÖNORM M 7132 by moisture 0% | 17.833 | 17.703 | 17.770 |
| Calorific value (MJ.kg$^{-1}$) determined by ÖNORM M 7132 in delivered condition | 9.736 | 7.811 | 6.660 |
| Calorific value (MJ.kg$^{-1}$) determined by STN ISO 1928 in delivered condition | 9.855 | 7.934 | 6.785 |
| Ash content (%) | 0.97 | 1.44 | 0.94 |
| | Pile 3 Sampling 5 February 2020 | | |
| Moisture content (%) | 60.2 | 58.9 | 60.5 |
| Combustion heat (MJ.kg$^{-1}$) | 19.207 | 18,198 | 18.457 |
| Calorific value (MJ.kg$^{-1}$) determined by ÖNORM M 7132 by moisture 0% | 17.865 | 16.856 | 17.115 |
| Calorific value (MJ.kg$^{-1}$) determined by ÖNORM M 7132 in delivered condition | 5.641 | 5.485 | 5.286 |
| Calorific value (MJ.kg$^{-1}$) determined by STN ISO 1928 in delivered condition | 5.768 | 5.611 | 5.413 |
| Ash content (%) | 0.93 | 0.65 | 0.61 |
| | Pile 4 Sampling 5 August 2020 | | |
| Moisture content (%) | 55.5 | 60.3 | 58.9 |
| Combustion heat (MJ.kg$^{-1}$) | 18.541 | 18.103 | 18.961 |
| Calorific value (MJ.kg$^{-1}$) determined by ÖNORM M 7132 by moisture 0% | 17.199 | 16.761 | 17.619 |
| Calorific value (MJ.kg$^{-1}$) determined by ÖNORM M 7132 in delivered condition | 6.309 | 5.183 | 5.806 |
| Calorific value (MJ.kg$^{-1}$) determined by STN ISO 1928 in delivered condition | 6.434 | 5.310 | 5.932 |
| Ash content (%) | 0.97 | 0.87 | 0.68 |

Logically, the chips from the top layer of the pile showed the highest moisture content, as they are most exposed to atmospheric conditions. On the contrary, the chips from the bottom of the pile were the most dried, as the energy-efficiency values show. Thus, the results also confirm the development of temperatures in the piles, where the highest temperatures during sampling were recorded in the lower part of the piles. The real increase in the moisture content at all height levels of the piles in correlation with the length of storage proves that the disproportionately long period of chip storage in uncovered piles (more than 1 year) significantly worsens their potential for energy use. With increasing storage times, all significant energy parameters also deteriorate. The maximum storage period for this type of wood chip can be considered to be half a year.

### 3.4. Fungal Activity in Experimental Piles

Overall, eight species/genera were identified in the samples taken from the first experimental pile (samples were taken on 6 February 2019)—Table 2. The most dangerous species for human health are species of the genera *Aspergillus* sp., *Mucor* sp., and *Penicillium* sp., as well as the mold *Cladosporium herbarum*. The highest numbers of colonies of these dangerous microorganisms were recorded in the sample from the upper part of the pile— 1.5 m ($18.7 \times 10^5$ CFU.g$^{-1}$), which confirms the increased danger for operators and workers handling chips. However, dangerous concentrations of spores of microorganisms were found at all levels of the first pile. However, due to the drier character of the weather in the storage period, the concentrations of microorganisms were still at a relatively high level.

**Table 2.** Identified fungal biological risks in storage piles.

| Pile Nr./Sampling Date | Sample Height | | |
|---|---|---|---|
| | Identified Microorganisms | | |
| | 0.5 m | 1.0 m | 1.5 m |
| Pile Nr. 1, sampling 6 February 2019 | *Penicillium citrinum; Penicillium* sp.; *Aspergillus brasiliensis; Mycelia sterilia; Mucor piriformis; Rhizomucor* sp. | *Penicillium citrinum; Penicillium* sp.; *Aspergillus brasiliensis; Mycelia sterilia; Cladosporium herbarum* | *Geotrichum candidum; Aspergillus brasiliensis; Penicillium* sp.; *Cladosporium herbarum; Mycelia sterilia; Mucor piriformis; Rhizomucor* sp. |
| Pile Nr. 2, sampling 12 August 2019 | *Penicillium expansum; Penicillium* sp.; *Mycelia sterilia; Paecilomyces* sp. | *Penicillium italicum; Penicillium expansum; Penicillium* sp.; *Mycelia sterilia* | *Penicillium expansum; Penicillium* sp.; *Mycelia sterilia* |
| Pile Nr. 3, sampling 5 February 2020 | *Aspergillus brasiliensis; Penicillium frequentans; Penicillium* sp.; *Mycelia sterilia* | *Aspergillus brasiliensis; Penicillium frequentans; Penicillium* sp.; *Mycelia sterilia* | *Aspergillus brasiliensis; Penicillium frequentans; Penicillium* sp.; *Aspergillus* sp.; *Mycelia sterilia* |
| Pile Nr. 4, sampling 5 August 2020 | *Penicillium diversum; Mycelia sterilia; Penicillium frequentans; Aspergillus flavus; Gliocladium catenulatum; Yeasts* | *Mycelia sterilia; Penicillium* sp.; *Gliocladium catenulatum; Penicillium expansum; Yeasts* | *Aspergillus brasiliensis; Penicillium expansum; Penicillium frequentans; Penicillium flavus; Yeasts* |

Overall four species/genera were identified in the samples taken from the second experimental pile (samples were taken on 12 August 2019)—Table 2. The most dangerous species for human health are species of the genera *Penicillium* sp. The highest numbers of colonies of these dangerous microorganisms were recorded in the sample from the bottom part of the pile—0.5 m ($14 \times 10^4$ CFU.g$^{-1}$), which confirms the fact that, when the material moisture content is higher than 40%, the development of phytopathogens is considerably slowed down. However, the reduction of health risks is also associated with the deterioration of the energy parameters of the samples. Naturally, it is also affected by weather conditions and the development of atmospheric temperatures.

Overall five species/genera were identified in the samples taken from the third experimental pile (samples were taken on 5 February 2020). The most dangerous species for human health are species of the genera *Aspergillus* sp. and *Penicillium* sp. The highest numbers of colonies of these dangerous microorganisms were recorded in the sample from the upper part of the pile—1.5 m ($9.8 \times 10^5$ CFU.g$^{-1}$). This has repeatedly confirmed the increased danger for operators and workers who handling chips as they handle the top layers of stored material. However, dangerous concentrations of spores of microorganisms were found at all height levels of the pile. Despite the relatively higher humidity and relatively favorable temperatures, however, the number of species identified and their total abundance did not exceed the values identified at first sampling after the first storage period.

Overall, 10 species/genera were identified in the samples taken from the fourth experimental pile (samples were taken on 5 August 2020). The most dangerous species for human health are species of the genera *Aspergillus* sp. and *Penicillium* sp. The highest numbers of colonies of these dangerous microorganisms were recorded in the sample from the bottom part of the pile—0.5 m ($15.4 \times 10^4$ CFU.g$^{-1}$), but very similar numbers

of microorganisms were also recorded in the middle part of the pile. This confirmed the danger of the stored material even after more than two years from its storage establishment. Dangerous concentrations of microorganism spores were still present at all height levels of the pile. Although the number of identified species exceeded the abundance at the first sampling, the concentration of spores was already slightly lower than after the first period.

From the point of view of the energy parameters of the stored biomass, its utilization is most suitable within half a year of storage. At the same time, however, the greatest health risk in this period is in terms of the concentration of phytopathogens of fungi and molds. However, this is largely influenced by the development of weather and atmospheric temperatures in a given storage period, which can have a relatively significant effect on these properties of the stored biomass.

## 4. Discussion

The results of the experiment confirmed that the energy properties of wood chips stored in small piles significantly deteriorate for more than half a year in connection with the extension of the storage period. After a year and a half of storage, the moisture content of the stored material almost doubled due to atmospheric conditions in the lower level of the pile. The calorific value, according to all laboratory methods, also decreased continuously with increasing storage time. In [45], the energy parameters of poplar wood chips were evaluated for a storage period of 6 months. As in the case of beech chips, the calorific values and moisture content improved during the first half of the year. The long-term storage of wood chips from birch (*Betula papyrifera*) in piles 3 m high for 13 months was also evaluated [25]. This research also confirmed that storing chips in this form of piles for more than a year will cause a deterioration in their energy properties and an increase in moisture content.

The results also showed that the length of the storage period did not have a significant effect on the abundance and risk of phytopathogen content in the stored piles. With the length of storage, only the number of identified harmful species of phytopathogens was changed and, at the same time, their number of colonies increased. Species of the genus *Aspergillus sp.* represented the most numerous phytopathogens in willow chip piles, as well as in a study from heating plants in Poland [46]. Similar findings were confirmed by a study by authors from Finland [47]. The results confirmed that spore concentrations above $10^4$ CFU.m$^{-3}$ can seriously endanger workers' health.

In [23], wood chips of spruce (*Picea abies*) and poplar (*Populus tremula*) were stored in piles of the same size (4 × 4 × 2 m) for 176 days. Temperatures in these piles reached maximum levels up to 50 °C, while the medians of temperatures at different height levels were mostly in the range of 20–30 °C. Similar to the research in this work, it was confirmed that storage in smaller piles can significantly affect the risk of spontaneous combustion. However, in addition to the dimensions, the temperatures in the piles are also affected by the input parameters of the chips from which the piles are based and also by the development of atmospheric conditions.

## 5. Conclusions

The work results confirmed that choosing the method of storing chips in smaller, spatially divided piles can lead to a significant minimization of the risk of spontaneous combustion and fire. In the wood chip pile with a total height of 4.0 m and volume of 163.5 tons were temperatures after a half-year of storage above 60 °C, permanently [24]. However, the positive effect of this method of chip storage has not been to minimize the production of phytopathogen spores and the associated risk to the health of workers handling it. The storage period of more than half a year worsens the energy properties of the stored wood chips. The benefits of a validated method for beech wood chips can only be defined in terms of reducing the risk of fire.

However, as the moisture content of the stored chips increases, a reduced exposure to wood dust during handling can be expected. This assumption is also confirmed by the results of the analysis of wood dust in thermally modified wood [48].

The production of spores of phytopathogens is a fundamental problem that can only be partially affected by the storage method. This is related to the biological nature of the stored material. Based on current knowledge, this risk factor can only influence the use of personal protective equipment by workers who handle the material, along with their current continuous training and motivation [49].

The applied method of this research would need to be verified in other conditions and on other types of wood species. However, we assume that the results will make it possible to generalize this storage method to minimize the risk of spontaneous combustion of wood chips. This storage method will need to be further verified and developed with other methodological approaches to minimize risks in the long-term storage of biomass.

**Author Contributions:** Conceptualization, M.G. and M.L.; methodology, M.L.; validation, M.G. and M.L.; formal analysis, M.L.; investigation, M.G.; resources, M.G. and M.L.; data curation, M.G.; writing—original draft preparation, M.G.; writing—review and editing, M.G. All authors have read and agreed to the published version of the manuscript.

**Funding:** This research was funded by the Slovak Research and Development Agency (grants number APVV-19-0612 and APVV-18-0520).

**Institutional Review Board Statement:** Not applicable.

**Informed Consent Statement:** Not applicable.

**Data Availability Statement:** Data sharing not applicable.

**Conflicts of Interest:** The authors declare no conflict of interest.

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
