# Peer review of "Wood Chip Storage in Small Scale Piles as a Tool to Eliminate Selected Risks"

_forests, doi:10.3390/f12030289_

Round 1
Reviewer 1 Report
This manuscript examines the impact of storage time on the calorific value and mold content of beech chips.
A more extensive literature review needs to be undertaken and put in the introduction to contextualize the work. The work cited focused on chip storage for energy use. However, there is a significant amount of work on chip storage for pulp and paper production that is not discussed. There are a lot of parallels between storing biomass for each of these applications. I encourage the authors to look at Swedish and Canadian work from the 1960s to 1980s on this topic.
The test piles were very small by industrial standards. The physical and biological phenomena that are observed in industrial scale chip piles cannot be easily replicated in smaller piles. The authors need to discuss this difference. In what ways is the current experiment not representative of an industrial scale pile? How might this impact the applicability of the results?
The word ‘dendromas’ is included a couple times. I’m not familiar with this word in English. I would recommend ‘biomass’ or ‘woody biomass’ instead.
Table 1. Some statistical analysis of the results is needed. In particular, is there a difference in calorific value at 0% MC? These values appear to be quite similar, but there is no indication of variability. It would be helpful to better distinguish to what extent the loss in calorific value is just a function of moisture content.
The authors refer to ‘dangerous’ microorganisms several times, yet this is not well substantiated. The microbial characterization is to the genus level in many cases. There is a great deal of variability between species within these genera. Some pose a human health concern. Some do not. Some molds can indeed be a human health concern, but this is not within the scope of the work. Better to just stick with what was observed and not speculate on potential health impacts without more thorough analysis.
The fungal data need to be tabulated or represented in a figure.
The conclusion states that the smaller piles reduce the risk of spontaneous combustion. This is probably true, but this can’t be concluded based on the data as no data were presented or cited from a larger pile.
Author Response
Responses in attached file

Reviewer 2 Report
Suggestions attached to the file.

Author Response
Responses in the attached file (responses in your comments).

Reviewer 3 Report
The manuscript is well written however the Introduction, Results and Discussion need improvements. I read the paper and could not conclude why the study was necessary.
Please authors justify your work, explaining why this paper is important for the community, how these results will impact the forest products industry.

Author Response
Responses in the attached file (responses directly in your comments).

Round 2
Reviewer 1 Report
Much improved.
Author Response
Thank you for precisely reviewing the manuscript.
Reviewer 2 Report
As the second round came in, I noticed that:
1) The major point of the article is still not clear. Is it a paper on biomass energy, or health risks?
2) Introduction is poor. It lacks sufficient context to the reader. Clear objectives, overall goal, rationale, and possible hypothesis. Is it short because of others papers published by the group?
3) More importantly: The authors claimed that the health risks on this paper is heavily associated with their previous work. So, what is the purpose of this paper?
4) Why Fagus sylvatica?
5) How many replicates per 6 months? It seems to me it was one. This is not sufficient to infer anything.
6) It is kind of obvious that a pile sitting for 2 years in the field will present degradation.
7) Aspergillus is the most common fungus found globally.
Author Response
Responses in attached file

Reviewer 3 Report
The authors made the changes that were suggested and the paper greatly improved.
Author Response

(The authors gave the same response as above.)
